# ABANICCO: A New Color Space for Multi-Label Pixel Classification and Color Analysis

**DOI:** 10.3390/s23063338

**Published:** 2023-03-22

**Authors:** Laura Nicolás-Sáenz, Agapito Ledezma, Javier Pascau, Arrate Muñoz-Barrutia

**Affiliations:** 1Departamento de Bioingeniería, Universidad Carlos III de Madrid, 28911 Leganes, Spain; lnicolas@ing.uc3m.es (L.N.-S.);; 2Instituto de Investigación Sanitaria Gregorio Marañón, 28007 Madrid, Spain; 3Departmento de Informática, Universidad Carlos III de Madrid, 28911 Leganes, Spain

**Keywords:** image color analysis, image analysis, semantics, fuzzy color space, color modeling, color segmentation, color classification, human perception

## Abstract

Classifying pixels according to color, and segmenting the respective areas, are necessary steps in any computer vision task that involves color images. The gap between human color perception, linguistic color terminology, and digital representation are the main challenges for developing methods that properly classify pixels based on color. To address these challenges, we propose a novel method combining geometric analysis, color theory, fuzzy color theory, and multi-label systems for the automatic classification of pixels into 12 conventional color categories, and the subsequent accurate description of each of the detected colors. This method presents a robust, unsupervised, and unbiased strategy for color naming, based on statistics and color theory. The proposed model, “ABANICCO” (AB ANgular Illustrative Classification of COlor), was evaluated through different experiments: its color detection, classification, and naming performance were assessed against the standardized ISCC–NBS color system; its usefulness for image segmentation was tested against state-of-the-art methods. This empirical evaluation provided evidence of ABANICCO’s accuracy in color analysis, showing how our proposed model offers a standardized, reliable, and understandable alternative for color naming that is recognizable by both humans and machines. Hence, ABANICCO can serve as a foundation for successfully addressing a myriad of challenges in various areas of computer vision, such as region characterization, histopathology analysis, fire detection, product quality prediction, object description, and hyperspectral imaging.

## 1. Introduction

Color plays a crucial role in computer vision: thus, color analysis is essential to tasks across various industries, such as medicine, satellite imaging [1], food quality assessment [2], visual scene description, and city planning. Despite its importance, color analysis remains a challenging problem, due to the complex nature of color, the lack of clear transitions or limits, the bias of human perception, the effects of context, and the semantic gap between subjective human perception and computer color representation.

Throughout history, there have been many attempts to deal with the complexities of color representation and analysis. Color has been represented in various ways, including the traditional color wheel and, in the age of computer vision, using color spaces, such as RGB and CIELAB. Computer vision is the branch of computer science that deals with the many challenges of color analysis and representation. A color vision algorithm should be able to reproduce the human experience of color perception, and to identify the colors present in an image, giving them a color name or color term. Additionally, the algorithm should be able to locate these color terms within a digital color space, and to use these subspaces for accurate color classification and image segmentation. However, several problems exist that render the creation of such a perfect color analysis algorithm difficult. The main problem with color vision is that human perception of color is not homogeneous: each person has their own experience of color, influenced by context, culture, education, and lighting [3,4]. Furthermore, human perception is also influenced by the phenomenon of color constancy, by which our brains perceive color homogeneously on surfaces, despite possible changes of lights or reflections of brighter colors, in contrast to the pixel-level differences in color detected by computers [5]. Thus, it is impossible to create a computer method able to reproduce the infinite possible experiences of color. The second problem, influenced by the differences in human perception of color, is that color names are not universal. The same shade of red could be called “crimson” or “cherry”, depending on the observer [6]; therefore, there is a great disparity between linguistic descriptions of color and computer representations. This problem is known in the literature as ’the semantic gap’ [7]. The ultimate challenge for color vision algorithms is that there are no well-defined boundaries between color categories, i.e., there are no immediate methods of finding, for example, the limit between orange and red, or between yellow and green. Humans will, in fact, resort to compound names (reddish-orange, yellowish-green) when confronted with colors located in fuzzy areas between more established terms [8].

Many alternatives have been introduced to deal with these problems, in order to create accurate computer vision methods for color detection and classification. The current research on color analysis primarily focuses on two areas: (1) color naming and detection; (2) color segmentation. Studies on color naming and detection aim to find a mapping between digital color representation and human language for color: this is a complex task, because of the subjective nature of human understanding of color [9]. Alternatively, fuzzy models, which handle the ill-defined boundaries between colors, are able to close the semantic gap while also performing well in the second area: color segmentation [8]. Another approach to color segmentation is color quantization, which is based on clustering, but it is limited by dependence on the training set [10,11].

In this paper, we introduce ABANICCO (AB ANgular Illustrative Classification of COlor). ABANICCO is a novel approach to color detection, naming, and segmentation that addresses the above limitations, and provides more accurate analysis of color. The proposed method uses color theory, fuzzy logic principles, and multi-label systems to create an efficient model that can perform the aforementioned tasks automatically, making it suitable for real-world applications. The main highlights of our method are:ABANICCO is an easy-to-use, automatic, non-supervised method for color analysis segmentation, classification, and naming;it is based on objective color theory, thus eliminating human color bias, and making the analysis trustworthy and reproducible;it incorporates fuzzy logic concepts to approximate color naming to human linguistics and understanding, thus bridging the semantic gap;finally, ABANICCO allows the user to make informed decisions, and to modify the obtained results, to fit specific applications: this is possible thanks to ABANICCO’s ease of use and excellent graphical representation of the detected colors.

The remainder of the paper is organized as follows: Section 2 introduces some fundamental concepts of color analysis, which are necessary in order to comprehend the problem—with a particular emphasis on those aspects that are most pertinent to the research proposed here; Section 3 presents the state of the art, and its limitations; in Section 4, we introduce ABANICCO and its innovations, with respect to the state-of-the-art methods; Section 5 relates the steps taken to build our model; Section 6 shows the experimentation with the proposed approaches, and their results; finally, Section 7 summarizes our work. Additionally, we include a Graphical Abstract and in Appendix A, extra technical information about the steps in the creation of ABANICCO.

## 2. Fundamentals

### 2.1. Color Theory

Color theory is the science that explains how humans perceive color: this perception results from how light is absorbed, reflected, and transmitted by the objects around us. When light strikes an object, some wavelengths are absorbed while others are reflected: this reflection is what allows us to see the color of an object. Nevertheless, all humans have different cultural and contextual associations with color that vary our perceptions [12,13,14]. As Newton said, “Colour is a sensation in the viewers’ mind” [15].

Three color-making attributes are used to communicate this subjective experience of color: hue; hue purity (expressed as colorfulness, chroma, saturation, or dullness); and lightness or darkness. In turn, hues are divided into primary, secondary, and tertiary. The primary hues are Red, Yellow, and Cyan. Blending two primary hues gives the secondary ones: Green, Orange, and Purple. Lastly, we obtain tertiary hues by mixing a primary hue with its nearest secondary. The six tertiary hues are Red-Orange, Yellow-Orange, Yellow-Green, Blue-Green, Blue-Violet, and Red-Violet. However, the nature of color is continuous; therefore, the best representation of colors is the color or hue wheel created by Sir Isaac Newton in 1704 [16]. Figure 1A shows how this hue wheel can be understood as the progressive blending of the primary, secondary, and tertiary colors represented inside it. It is important to note that some colors are not represented in the color wheel, because the color wheel only shows pure or fully chromatic colors. Modifications to these pure hues create the rest of the possible colors. The hues can be modified by changing the color-making attributes of purity and lightness with achromatic or neutral colors—white, black, and gray—to obtain tints, shades, and tones, respectively. A wider variety of colors can be seen in the tints, shades, and tones images of Figure 1B. This difference between pure hues and modifications can be better understood if we single out certain shades and tones (Figure 1D). Triangle 5.t shows what can be referred to as “forest green”; however, this deep shade of green is usually described simply as “dark green”. Circles 1 to 4 show increasingly dark (1–4.s) shades and duller tones (1–4.t) of orange: when observed separately, it is clear that these colors are what we usually refer to as “brown.” Brown is technically a near-neutral dark-valued orange; however, in contrast to dark green, almost nobody would refer to brown as dark orange: this is a crucial point to understand, because most research conducted to date on color detection and classification aims to find a unique space for brown as a pure hue instead of treating it as a modified hue, the same as dark green or dark blue [17,18,19]. This repeated inaccuracy in color research—which contradicts even Goethe’s and Newton’s precise placement of brown as dark orange in their color wheels [20,21]—likely stems from the complicated gap between color theory, color perception, and color naming [22].

### 2.2. Color Terms

The continuous nature of color, with no clear transitions or limits, makes classifying and naming individual colors problematic. The first attempts at color naming and classification were driven by the boom of exploratory trips and expeditions in the 18th and 19th centuries ([18,19,23,24,25]); however, all these proposals relied heavily on context and cultural cues. In the 20th century, two significant attempts were made at color naming standardization, and remain the baseline against which most research is compared:To standardize color naming, the Inter-Society Color Council (ISCC) and the National Bureau of Standards (NBS) first established, in the 1930s, the ISCC–NBS System of Color Designation [26], which defines three levels of color naming: Level 1 consists of 10 color terms; Level 2 adds 16 intermediate categories; and Level 3 uses brightness and pureness modifiers to obtain 267 color categories. This system was used in one of our experiments, to test the accuracy of ABANICCO’s color naming in Section 6.1.In 1969, Berlin and Kay examined the effect of culture on the differences in color naming in natural languages. They found 11 basic color categories in most of them: white, gray, black, yellow, orange, red, pink, purple, blue, green, and brown. In their work *Basic colour Terms: Their Universality and Evolution* [27], they were able to link the universality of these terms to similar evolutionary processes. They also found that within these categories, the central regions would elicit higher consensus than the boundaries, where consensus tapered off, consistent with the continuous nature of color.

Berlin and Kay’s categorization and the ISCC–NBS System are the most-used color terms standardizations in digital imaging and color research nowadays. However, both present approaches have problems that limit their practical use. Berlin and Kay’s color terms fail to combine successfully to create secondary and intersectional classes. Additionally, the experiments were conducted using exclusively high chroma levels, never testing less saturated colors. Furthermore, the ISCC–NBS color terms may seem arbitrary to different users who do not agree with the provided descriptions. Moreover, the terms on Levels 2 and 3 do not follow logical reasoning that could make them understandable to any human or machine, as demonstrated by Anthony E. Moss in [28].

### 2.3. Color Models

Due to the difficulties in finding appropriate, accordant names for colors, color models were introduced throughout the 20th century. Color models are systems that describe colors using numerical values instead of linguistic terms. The most influential one is the Munsell color system, created in 1913 by Professor Albert H. Munsell as a rational way to describe colors [29]. The fundamental premise of the Munsell color system is the distribution of colors along three perceptually uniform dimensions: hue; value; and chroma. The placement of the colors was decided by measuring human responses, instead of trying to fit a predefined shape, thus obtaining an irregularly shaped cylindrical space that avoids distorting color relations [30]: in this space, hue is measured in degrees around horizontal circles, chroma goes radially outwards from the neutral vertical axis, and the value increases vertically from 0 (black) to 10 (white). The irregular shape of the Munsell color space allows for wider ranges of chroma for specific colors, which follows the physics of color stimuli and the nature of the human eye. This non-digital color system is the most-used in psychology and soil research [31,32].

With the rise of computer operations, digital color models were created to suit the new tasks better. The main digital color spaces are RGB, HSL/HSV, and CIELAB. A visual representation of these can be seen in Figure 1C. RGB was created in 1996 for internet and display purposes: in this space, colors are represented as three coordinates corresponding to red, green, and blue values. Despite being the most widely used color space in digital applications, the colors it can represent are limited, device-dependent, and discordant from human perception and interpretation. By contrast, HSL/HSV spaces were designed circularly, to better align with human perception of color-making attributes: these spaces consist of a three-dimensional cylindrical transformation of the RGB primaries into a polar coordinate system, obtaining hue, saturation, and value (in HSV) or lightness (in HSL). Finally, CIELAB is a uniform color space, based on human perception, that achieves the best compromise between complexity and accurate color representation by adopting the concepts of the Munsell color system. In CIELAB, the geometrical distance between any two points represents the perceived color differences without forcing the color relationships into strict geometrical shapes: this makes CIELAB more perceptually linear and decorrelated than the alternatives [33,34,35].

## 3. Related Work

Color analysis poses numerous challenges, including variations in perception, limited linguistic mapping between language and machines, and lack of clear boundaries, as explained in the previous section. Currently, color analysis research primarily focuses on resolving two related problems: (1) color naming and detection; and (2) color segmentation.

Much research has been devoted to color naming and detection, with the aim of finding a suitable mapping between the digital representation of colors and the language employed by humans when referring to them to bridge the semantic gap [6,9,36]. However, this is a complex task due to the subjective nature of human understanding of color, which can vary, based on factors like age, context, profession, and eye morphology. The only way to truthfully close the semantic gap would be to run a multitudinary experiment with a sufficiently heterogeneous cohort that would consider every possible experience of color: however, this would be highly impractical.

Fuzzy models, which tackle the notoriously ill-defined boundaries between colors, are being used as a practical alternative. Fuzzy set theory is closer to the human cognitive process of color categorization than crisp color partitions [7]: because of this, several research works have applied fuzzy logic to successfully obtaining a correspondence between human perception, color language, and computational color representation. Within this approach, three subgroups of methods stand out: (1) those that rely on geometrically regular membership functions; [2,37,38]; (2) those that formalize fuzzy color spaces from human-obtained prototypes [6,36]; and (3) those that create adapted fuzzy color spaces dependent on application [8,39,40,41,42]. Methods in the first subgroup are problematic, because they rely on color spaces with arbitrary distances between color categories. Moreover, using regular, homogeneous functions disregards the irregular form of color terms [43]. Methods in the second subgroup provide good color models, but depend on the number of study participants, and focus on a limited number of color terms obtained from subsets of a single cultural environment (like Berlin and Kay). Methods in the third subgroup favorably address the limitations of the previous methods, and achieve good results in color segmentation, as well as in color detection and naming: however, the resulting pipelines are complex and inaccessible for regular use, and are challenging to adapt to new applications.

In addition to fuzzy color models, color quantization is widely used for color segmentation. Color space quantization is based on clustering, usually by nearest neighbor classification or color thresholding. Classification based on nearest neighbors finds the *k* closest neighbors to any given pixel in a set of predefined color classes [10,11]: this approach is limited by the necessary definition of the clusters and boundaries, which depends entirely on the training set, although some recent research provides more robust results [44]. Alternatively, color thresholding divides the space into segments defined by mostly-linear boundaries that need to be manually decided: this step has been solved using machine learning techniques, which require a priori knowledge of the labels of the color categories and training process, or by complex deep learning optimization algorithms [45,46,47]. Like the previous method, this approach is severely limited by the dependence on training data [10,11].

Nevertheless, the problem most previous methods have in common is disregarding color theory fundamentals. Excepting the most recent fuzzy model approaches, most of the above-mentioned approaches are based on misinformed notions of what colors are: these inaccuracies are often due to the confusion between pure hues and color terms. Basic Color Terms (BCTs, as introduced by Sturges & Whitfield in 1991) include 11 categories: Black; White; Red; Green; Yellow; Blue; Brown; Orange; Pink; Purple; and Gray [27,43]. Since then, many research works have aimed to locate these color terms in the color spaces, as if they were pure hues: this becomes particularly problematic in the case of brown [36], for the reasons stated in the previous section.

## 4. Proposal

This paper introduces ABANICCO, a new approach to pixel classification, color segmentation, and naming. Our proposal boasts groundbreaking innovations in the field. Firstly, we have formalized the notion of our modified AB polar, two-dimensional color space as the primary object for color analysis. This innovative representation offers a clear graphical understanding of the colors present in an image, and of how they are organized. Our modified color space is divided into 12 categories: Pink; Red; Red-Orange; Yellow-Orange; Yellow; Green; Teal; Blue; Ultramarine; Purple; Brown; and Neutral. These categories were determined by color theory, thus avoiding the need for tedious experiments and a large number of experiment subjects.

Secondly, we present a methodology that utilizes fuzzy principles and multi-label systems to name the identified colors, bridging the semantic gap by building on fuzzy color models. The functions used are based on the color categories established through color theory, and consider irregular color shapes in line with concepts outlined in prior research [29,43].

As a third innovation, our color mapping allows for accurate color segmentation. As we do not rely on human-made labels or annotations, our color segmentation is objective; moreover, it avoids problems such as the effect of light, issues with perception, the phenomenon of color continuity, and the effects of context and personal bias.

These innovations are visually summarized in the graphical abstract included.

Finally, we propose a simple pipeline for image segmentation, based on the color mapping provided by ABANICCO. The main advantage of this pipeline over existing approaches is the possibility of the user making a fully informed decision about the areas to segment, thanks to the polar description ABANICCO provides of the images.

## 5. Methods

To create the proposed method, we approached the problem of color analysis in two steps. Firstly, we solved the problem of automatic color identification and segmentation, by using color theory and simple geometry concepts to study the color wheel, and to obtain the pure hue bases in the CIELAB color space. We named this step “Geometric knowledge generation.” Secondly, we used fuzzy logic notions, color theory and multi-label classification to name and describe the identified colors: this step was called “Fuzzy space definition and multi-label classification for colour naming and description”.

### 5.1. Geometric Knowledge Generation

The purpose of this step was to create a simplified color space for color classification and naming. We chose to work with CIELAB color space, due to its advantageous properties, as explained in Section 2.3.

The first task was to find the exact location of the pure hues in the color wheel within the CIELAB color space. As color space dimensions are decorrelated, the luminance component is unnecessary for understanding a given color’s chromatic content: this meant that we could discard the luminance dimension, and work on the reduced AB two-dimensional space. To find the location of the main crisp or pure hues in this reduced color space, we selected an example of a complete color wheel, including hue modifications. We calculated the double AB histogram, showing the location of the crisp hues. The resulting geometrical distribution radiated from a center point, as shown in Figure 2A. The skeletonization of the double histogram provided endpoints and branches following a circular path. Due to their disposition, the lines between the center of the dual histogram and each endpoint could be used to represent the bases of the pure chromogens. Figure A1 provides a better look at this representation.

Once we knew where the pure bases were, the next task was to find the boundaries between them. We considered two options for locating the boundaries between the crisp color categories: angle bisectors and classification methods. Both were tried out before moving to the next step, to decide on the best approach. The classification methods were more complex to use, needed training, and took longer to provide results, and the obtained classifications were unsatisfactory, both in resulting boundaries and classes created. An example of these unsatisfactory boundaries applied to natural images was the obtained segmentations of a gradient of colors in Figure A2 and Figure A3. We observed how the machine learning classifiers failed to create appropriate boundaries between the three present colors, each generating a single class of arbitrary length instead of the expected classes of set length obtained by the bisectors approach: these results, together with the simplicity and efficiency of the bisectors approach, explain why we chose it over other classification methods. The experiments carried out to determine the best method are described in Appendix A, and the obtained segmentation of the whole range of hues in Figure A2 and Figure A3.

Using angle bisectors, we obtained precise color regions that, for simplicity, were then grouped according to color perception, shape, CIELAB construction, and the Berlin and Kay theories [27,33], in 10 groups: Green; Yellow; Light Orange; Deep Orange; Red; Pink; Purple; Ultramarine; Blue; and Teal.

It is important to note that neutrals (black, white, and gray) and near-neutral brown were not present in this classification, as they are not pure chromatic colors. In the CIELAB color space, the pure hues are located far from the center, and are radially distributed. All modifications of these hues then fall gradually closer to the center, with the shades having lower luminance than the tints, and the achromatic colors falling in a gradient aligned on the L axis. This can be seen in the CIELAB diagram of Figure 1C. In our reduced AB space, this meant that the neutral colors, black, white, and gray were located in the center of the space. The shades and tones of orange, commonly named brown, were located close to the center, within the red, orange, and yellow areas. Figure 2 shows how this area corresponded to the lower left corner of the upper right quadrant in the AB space (consistent with the mixed range of yellows and reds), close to the L axis (due to their near-achromacity); therefore, brown shades could be aligned with any pure chromogen bases between red and warm yellow, depending on the angle and distance from the center.

The final task was the incorporation of browns and neutral as two extra classes. For this purpose, we included two additional areas in our simplified color space, which depended on the distance from the center: the first area was the circular area of radius $r_{1}$ (Critical Achromatic Radius) corresponding to the neutral pixels centered on the L axis—the whites, grays, and blacks in the image; the second area was a quadrant of radius $r_{2}$ (Critical Near-Achromatic Radius) corresponding to the positive values of A and B in the 2D color space, and comprising all shades of brown. The radii $r_{1}$ and $r_{2}$ corresponded to the first and second sets of branching points in the skeletonized double histogram, as seen in Figure A1 of Appendix A.

In this manner, a new, simplified color space was created, capable of classifying colors into 12 categories: Pink; Red; Red-Orange; Yellow-Orange; Yellow; Green; Teal; Blue; Ultramarine; Purple; Brown; and Achromatic, as seen in Figure 2. Within our pipeline, this new color space was used in the pixel classification section: the RGB input image was converted into CIELAB, and the AB values were translated into polar coordinates (radius (*r*) and angle (θ)). Each pixel was then classified into one of the 12 categories with these coordinates. The categories were consistent with the unbroken areas obtained using angle bisectors over the color bases. This classification depended solely on the angle for the first 10 categories, the radius for the neutral category, and the angle and radius for the Brown category.

### 5.2. Fuzzy Color Space Formalization and Color Naming

As described in the previous section, using the angle bisectors on the double AB histogram, we were able to successfully detect and separate different chromogens, accurately segmenting the detected colors. In this Section, to approach the goal of defining an unbiased continuous space for color naming and classification, we applied the principles of fuzzy models applied to multi-labeling systems: in this way, we used trapezoidal heterogeneous membership functions to represent fuzzy colors. Namely, we defined areas of absolute chromogen certainty as the points lying on top of the previously established bases. Moving away from the pure bases, the membership degree to them decreased as we defined increasing gradients of membership for the colors on both sides. The approach was inspired by [8,39,40,41,42,48,49].

We defined the set of membership functions using polar coordinates, as shown in Figure 1B, and trapezoidal piecewise polynomials, to maintain continuity, simplifying the problem and matching the geometric angular analysis. For the angular membership, linear piecewise functions were fitted for each color class, so that they would output 1 when the angle of the pixel was in the middle (or peak) of the color class, 0.5 in the boundaries, and 0 when over the peak of the next or previous color class, as given by
(1)$$f\left(\theta ,a,b,c,d,e,g\right)=\left\{\begin{array}{cc} 0 & \;\mathrm{if}\;\theta \leq a,\\ \frac{\theta \;-\;a}{2(b\;-\;a)} & \;\mathrm{if}\;a<\theta <b,\\ \frac{\theta \;-\;2b\;+\;c}{2(c\;-\;b)} & \;\mathrm{if}\;b<\theta <c,\\ 1 & \;\mathrm{if}\;c<\theta <d,\\ \frac{\theta \;-\;2e\;+\;d}{2(d\;-\;e)} & \;\mathrm{if}\;d<\theta <e,\\ \frac{\theta \;-\;g}{2(e\;-\;g)} & \;\mathrm{if}\;e<\theta <g,\\ 0 & \;\mathrm{if}\;\theta \geq g, \end{array}\right.$$
where *a* was the peak for the previous color class, *b* and *e* were the angles corresponding to the boundaries, *c* and *d* were the angles adjacent to the peak of that color class, and *g* corresponded to the peak of the next color class.

This was done using the geometric knowledge learned from the AB reduced space. The radius membership functions were created to differentiate between achromatic (scale of grays), near-achromatic (brown), and pure colors. The function defining the achromatic area was modeled with two linear splines, as given in
(2)$$f_{Achromatic}\left(r,r_{1}\right)=\left\{\begin{array}{cc} 1 & \mathrm{if}\;r\;\leq \;r_{1},\\ 0 & \mathrm{if}\;r\;>\;r_{1}. \end{array}\right.$$
while three linear splines were used to define the near-achromatic area:(3)$$f_{Near-Achromatic}\left(r,r_{1},{r_{2}}^{\prime },r_{3}\right)=\left\{\begin{array}{cc} 0 & \mathrm{if}\;r\;\leq \;r_{1},\\ 1 & \mathrm{if}\;r_{1}<r<{r_{2}}^{\prime },\\ \frac{r\;-\;r_{3}}{{r_{2}}^{\prime }\;-\;r_{3}} & \mathrm{if}\;{r_{2}}^{\prime }<r<r_{3},\\ 0 & \mathrm{if}\;r\geq r_{3}, \end{array}\right.$$
and the chromatic area:(4)$$f_{Chromatic}\left(r,{r_{2}}^{\prime },r_{3}\right)=\left\{\begin{array}{cc} 0 & \mathrm{if}\;r\;\leq \;{r_{2}}^{\prime },\\ \frac{r\;-\;{r_{2}}^{\prime }}{{r_{2}}^{\prime }\;-\;r_{3}} & \mathrm{if}\;{r_{2}}^{\prime }<r<r_{3},\\ 1 & \mathrm{if}\;r\;\geq \;r_{3}. \end{array}\right.$$

In the above equations, $r_{1}$ is the Critical Achromatic Radius, ${r_{2}}^{\prime }$ is the Critical Near-Achromatic Radius, and $r_{3}$ is the Modified Critical Chromatic Radius, obtained from the skeletonized histogram.

It can be seen from Figure 2 that, while the splines for the near-achromatic and achromatic areas showed a similar logic to the angular membership functions—with a clear gradient between the two classes—that was not the case for the achromatic area. To account for this, the Critical Achromatic Radius ($r_{1}$) was used, as defined in the previous section: all colors with radii lower than $r_{1}$ were classified as achromatic. However, the Critical Near-Achromatic Radius ($r_{2}$) was used as the point at which the membership function would output 0.5 for near-achromatic and chromatic. Thus, two new points were created: the Modified Critical Near-Achromatic Radius (${r_{2}}^{\prime }$) and the Critical Chromatic Radius ($r_{3}$), both separated 5 points from the original Critical Near-Achromatic Radius ($r_{2}$).

With this approach, we obtained an exhaustive analysis of each shade in the input image. We obtained a list of the 10 main shades from each color category. Each shade was then described as a percentual combination of the main color categories required to create it. Thus, a shade with polar coordinates R = 41, Theta = 8° was named “44% Brown, 36% Light Orange, 20% Yellow”, corresponding to a yellowish light beige. We believe that this naming technique, based on mathematical principles, provided an excellent solution to the semantic gap, representing an optimal halfway point between human understanding and computer representation of colors.

To summarize, we made use of concepts from different disciplines in the development of ABANICCO: firstly, we centered our model on color theory, which we used to find the location of colors in space, and to represent and name them accordingly; then, we employed geometric analysis to find the best boundaries between color bases; finally, we combined fuzzy theory and multi-label classification, to create a continuous space. The resulting space was closer to human understanding of color, from which non-exclusive labels could be assigned to each detected color, to name them accurately, thus bridging the semantic gap.

## 6. Results and Discussion

In this section, we describe the experiments performed to validate our approach. This empirical evaluation provides a qualitative and quantitative verification of ABANICCO’s effectiveness beyond providing only a simple visual assessment of its results.

While color analysis is a crucial component in various computer vision tasks, the assessment of analytic models is decidedly troublesome. As explained in Section 2.1, the perception of color is influenced by many subjective factors, such as the individual’s age, gender, culture, and personal experiences. While most computer vision algorithms are assessed using human-made labels, in the case of color analysis, these human-made annotations can never be objective: they differ, depending on the group creating them. In addition, color measurement devices can be influenced by factors such as lighting conditions, substrate color, and sample size, making it difficult to achieve consistent and reliable results. Moreover, in most applications, color is often considered an attribute of an object rather than a standalone concept: this distinction, along with the challenges posed by human perception, context, and interpretation, results in a scarcity of ground truth data and adequate analysis metrics for color analysis as an individual concept.

To address these difficulties, we designed two types of experiment. The first type of experiment quantitatively evaluated ABANICCO’s accuracy in color classification and color naming, using as a benchmark the ISCC–NBS color classification Level 2 (presented in Section 6.1). The second class of experiment explored the use of ABANICCO in color-based image segmentation (presented in Section 6.2). We firstly conducted a qualitative evaluation of the performance of ABANICCO’s clustering-based color segmentation pipeline against the state-of-the-art fuzzy-based color segmentation algorithm (presented in Section 6.2.1 and Section 6.2.2); then, we quantitatively assessed the validity of this pipeline for object segmentation, using the ITTP-Close dataset (presented in Section 6.2.3). Thanks to this empirical evaluation, we provided a numerical comparison of the proposed model against state-of-the-art benchmarks.

### 6.1. Classification and Naming

To numerically measure ABANICCO’s accuracy for the tasks of color classification and naming, we applied our model to the official ISCC–NBS color classification Level 2 [26], established in 1955 by the Inter-Society Color Council and the National Bureau of Standards. This second level of the ISSCC–NBS color system includes a set of 29 colors made up of 10 hues, three neutrals, and 16 compound color names.

We fed our model the image seen in Figure 3A, where the 29 Level 2 ISSCC–NBS prototypes were represented as ordered squares of the same size. ABANICCO classified each of the colors into its 12 categories (Figure 3B), and provided a clear name for each of the individually detected colors. Figure 3 and Table 1 summarize the model’s classification and naming performance. Specifically, Figure 3 reflects the results regarding color classification, while Table 1 reports the naming performance of ABANICCO. This first experiment proved that our method could successfully classify the shades into the expected categories, and provide complete, coherent labeling for the ISCC–NBS system names. However, some of the terms in the ISCC–NBS system were incongruous from a color-wheel perspective [28]: for example, “Yellowish pink” and “Reddish Purple”, which combined two non-adjoining categories, skipping the middle “Red” and “Pink” sections. Our method solved these nonlinearities by classifying those colors within the skipped middle categories. These two scenarios (marked in bold in Table 1) were the only ones reporting discrepancies between our method and the standardized ISCC–NBS system. Thus, the quantitative evaluation of the segmentation and classification accuracies of ABANICCO told us that, initially, ABANICCO had 93.1% classification accuracy, which increased to 100% when we disregarded the mentioned nonlinearities.

### 6.2. Segmentation

Following the validation of ABANICCO’s accuracy in classification and naming, we assessed the effectiveness of our model’s color segmentation capabilities. We performed two sets of experiments: Section 6.2.1 and Section 6.2.2 focused on color segmentation, and Section 6.2.3 focused on image segmentation. As noted earlier, ABANICCO segments color without human intervention, thereby eliminating the possibility of human bias in color identification: however, this meant that a reliable color segmentation ground truth for comparison was unattainable. The available ground truths were created for image segmentation, and took color as an attribute of the objects in the image. These supervised methods for the segmentation of color as an attribute were subject to the influence of perception and preconceived notions of those objects. By contrast, considering color as a complete entity meant that each pixel held significance, and a more exhaustive range of “perceived colors” could be identified.

This distinction came to light in our second experiment (Section 6.2.1), in which we assessed the color segmentation abilities of ABANICCO, using an image of the flag of the United States of America. Flags are often used to test color segmentation algorithms, because they were originally designed using a limited number of easily recognizable and contrasting shades: these characteristics make it relatively straightforward to create a ground truth from images of flags. Per contra, this simplicity introduces a high context bias, as the annotator knows what they are looking for, and what the annotations are expected to look like—in this case, whole sections of one color. However, in natural images, variations in lighting and movement can alter the original hues, saturation, and intensity of the expected colors: for example, a red flag section may exhibit brown regions, while white areas will reflect the vibrant colors of neighboring sections—variations that are rarely reflected in human annotations. In fact, in our fourth experiment (Section 6.2.3), we explored how similar variations can occur with 3D objects of vibrant colors.

A color-analysis-focused method, such as ABANICCO, will detect these discrepancies in individual pixels, as the purpose is not accurate object segmentation; by contrast, methods built for object segmentation, based on annotations of colors as attributes, will not take notice of these variations. The particular task will thus define the adequacy of the method. Nevertheless, in order to provide a quantitative comparison of ABANICCO to the state-of-the-art methods, we incorporated a simple step of class remodeling into the output provided by ABANICCO, resulting in accurate image segmentation.

#### 6.2.1. Comparison with Fuzzy Color Spaces—Discrete Colors

For this second experiment, we compared ABANICCO with the most representative method of the state of the art for color classification. In their work, M. Mengíbar-Rodrigez et al. [42] used Fuzzy Logic and space optimization to obtain a suitable set of prototypes upon which to build a fuzzy color space for each image or set of similar images.

In contrast to ABANICCO, their method required annotations, and did not detect individual colors. Furthermore, in some cases, the user had to decide the number of categories for prototype learning. Their method returned, as output, a map of the membership degree of each pixel to the studied category, which they represented by modifying the rate of transparency of the segmented images. Specifically, they successfully segmented images of country flags into classes corresponding to each flag’s original colors. Compared to a human-generated ground truth of 12,000 color samples, they obtained a 0.955 f-score. The ground truth was not created as a complete segmentation of all the areas in the images, but rather as small, 10-pixel samples of consistent colors within the images: this approach prevented the possible variations of the expected colors from being included in the ground truth. Thus, their method was tested against a partial and biased truth.

In this section, we quantitatively compared the segmentation given by ABANICCO directly to their results, as opposed to using their ground truth, as it is not publicly available. The assessment of segmentation performance is most often determined through the use of quantitative metrics, such as object overlap and distance analysis: however, we argue that such approaches would fail to fully capture the nuances inherent in the membership maps obtained in [41]. For lack of an adequate, problem-aware, quantitative metric that could compare such different outputs [50], we preferred to assess the differences in performance quantitatively in Figure 4.

We compared the performance of both methods, using the image of the United States of America flag seen in Figure 4A. The image selected presented clear boundaries between colors. Moreover, the colors were arranged in clusters far away from each other in the color wheel, which can be appreciated in the polar description of the image in Figure 4G: this image is a prime example of how lighting and movement can affect the colors of an object, creating browns on the shaded areas of the red stripes, and new tones of blue as a result of dark-blue areas reflecting over the white stripes and stars. Nevertheless, the phenomenon of color constancy, human bias, and pre-existing notions about the flag will lead most observers to detect only three colors. These variations of the expected colors are marked in Figure 4, rows E and D.

As ABANICCO is designed for color segmentation and classification, its initial segmentation may not align with the ground truth, as it prioritizes accurate color recognition over object segmentation. In this case, it was clear that the segmentation errors were actually a result of ABANICCO’s focus on legitimate color classification: the areas in which either blue or red were reflected onto the white stripes were classified as blue and red, their true colors. The actual color of these areas can be seen marked with an orange and green star, respectively, in the full colors summary of Figure 4E. Therefore, the segmentation using ABANICCO was accurate in terms of color but not in terms of object edges or expected human segmentation. To achieve the expected segmentation, we used the information provided by the shade classification (Figure 4E ) and polar description (Figure 4G) to easily modify the class boundaries and classes, so as to best match the application (Figure 4F): this simple step of boundary remodeling allowed us to obtain the expected (biased) segmentation for this application, by controlled, informed decisions based on color theory. When we compare our modified segmentation with the one obtained using [41], we can see that the baseline method had problems classifying the different shades of blue, and also the red color, when it appeared darker due to lighting and movement (Figure 4H). This was most patent in the upper right corner of the image. The use of ABANICCO with an added class remodeling step outperformed the benchmark method in segmenting the expected red, blue, and white areas of the flag.

Thus, our method can easily go from pure color classification to contextual image segmentation, by allowing the user to make informed decisions concerning class boundaries.

In Section 6.2.3, we further explored ABANICCO’s capabilities for the problem of object segmentation, performing in-depth quantitative analysis for this specific problem.

#### 6.2.2. Comparison with Fuzzy Color Spaces—Continuous Colors

The image used in the previous experiment is easy to segment for clustering-based methods such as [41], due to the limited number of colors present, and their distinctive distribution within the color wheel; moreover, obtaining a ground truth of the expected categories within it is relatively easy: to address this, we repeated the evaluation performed in Section 6.2.1 on an image with no clear edges: the whole spectrum of hues of the color wheel, and different shades and tones of said hues. The image in Figure 5A is an excellent example of how the absence of precise edges and context impedes the creation of accurate ground truths for the analysis of true color segmentation. A human annotator would have to decide the number of classes of color present in the image, the exact boundaries between these classes, and whether the differences in value and saturation would affect these limits, thus making the ground truth dependent on the individual. As explained in the previous section, due to the lack of a precise measurement methodology for color segmentation, especially in images with no clear edges, the comparison between the two models in this experiment was limited to a visual assessment.

We applied the method presented in [41] and ABANICCO to the image shown in Figure 5A: the comparison between both models’ results is shown in Figure 5B,C. In this scenario, which presented a more challenging image, our method successfully identified and classified the existing colors into the 12 categories created according to color theory. By contrast, the method proposed in [41], without previous knowledge of the image, failed to create coherent categories, mixing blues with browns, and oranges with greens.

#### 6.2.3. Object Segmentation—ITTP-Close Dataset

In this final experiment, we present a quantitative analysis of ABANICCO’S performance when applied to the problem of object segmentation. To this end, the technique of class remodeling described in Section 6.2.1 was tested on natural images of the ITTP-Close dataset introduced in [51]: this dataset consisted of five images (450 × 500 pixels) of scenes with a close light source. All the images contained volumetric objects, some of which included highlights. The annotations for the ground truth were done by splitting the images into small subregions with guaranteed color constancy, and manually merging them when the colors were indistinguishable by the human eye.

Figure 6 shows the images in the dataset, and their corresponding ground truth; moreover, it presents an excellent example of why image segmentation based on color as an attribute is not the same as color segmentation. In Figure 6C, we see that in a singular area of the image, as defined by the ground truth, there was a wide range of distinct colors that even differed in hue: for example, the book appeared as mostly uniform, unsaturated greenish blue; however, it also showed pinkish browns from the reflection of the orange piece, and yellowish greens from the reflection of the yellow cylinder. These variations were not considered in the ground truth, as they would have interfered with object segmentation, which shows why using color for image segmentation is not the same as color segmentation.

In Figure 7, we see the polar description of this same image, as provided by ABANICCO. It is interesting to observe that in our polar AB color space, the different colors of an area appear naturally grouped in defined clusters; therefore, by simply moving the class boundaries, we obtain a perfect segmentation of the objects present in the image.

In this experiment, we were testing segmentation of objects with clear boundaries, which were easier to annotate than the previous cases explored in this work. Thus, we could easily test this segmentation pipeline against the method proposed by the authors of the dataset in [51]. In their work, they proposed a region adjacency graph framework with color space projective transform as a pre-processing technique. This method is not automatic, and requires initialization. In their work, the authors used the dataset median Intersection over Union score (dataset-mIoU) for the evaluation of their segmentation, as defined by
(5)$$\;\mathrm{dataset}-\mathrm{mIoU}\;=\sum_{k=0}^{K}\min\left(\mathrm{IoU}\left(S_{k}^{*},{\tilde{S}}_{k}\right),0.5\right),$$
where *K* was a number of ground truth segments. The quality range was also in [0,1].

The authors of [51] reported a dataset-mIoU of 0.85 for their method. By contrast, our segmentation pipeline achieved a dataset-mIoU score of 0.96, representing a 13% improvement.

### 6.3. Discussion

In Section 6, we conducted an empirical evaluation to assess the color naming, classification, and segmentation performance of our proposed method, ABANICCO, compared to different state-of-the-art methodologies. The reported results show that ABANICCO outperforms the benchmark methods in accuracy and simplicity.

Specifically, the performed experiments can be divided into two different categories: the first category dealt with color classification and naming, while the second dealt with color segmentation. Firstly, we proved that ABANICCO was able to correctly classify the colors on the official ISCC–NBS color classification Level 2. Moreover, for each color, ABANICCO provided a coherent name that was understandable for humans and machines alike, thus solving the inconsistency between human semantics and machine-extracted features—that is, the semantic gap. Secondly, regarding color segmentation, three additional experiments compared ABANICCO’s capabilities with state-of-the-art methods: these experiments showed that ABANICCO is the only method that prioritizes color analysis over area or object segmentation. Moreover, the proposed AB polar description of the colors allows the user to create clusters that solve image segmentation tasks easily. With this simple step of class remodeling, our segmentation pipeline achieved improved segmentation performance compared to state-of-the-art methods for object and area segmentation.

Thus, ABANICCO successfully dealt with the common pitfalls of other color analysis approaches. By basing our model on objective color theory, we avoided the danger of subjectivity, inability to handle variations in lighting and noise, and the lack of standardization of other models. Moreover, we ensured the maximum range of detectable colors, by building ABANICCO from CIELAB color space. Finally, ABANICCO is the least complex model compared to the state of the art. This, together with not needing any annotations, makes ABANICCO practical for a wide range of applications, and accessible to researchers, regardless of their field.

The experiments conducted suggest that ABANICCO could be useful in scenarios that involve changing objects, an unknown number of classes, limited data, unclear boundaries, or when the user lacks prior knowledge of the image or color theory. Moreover, its robustness and unsupervised nature make ABANICCO particularly useful in cases where deep learning techniques are unavailable, inefficient, or when labeled data is scarce or human-biased. The non-supervised character of the proposed method also eliminates the risk of human perception and context-driven errors in color analysis. Furthermore, ABANICCO’s potential extends beyond standalone use, and could serve as an initial step in more complex deep learning tasks, such as reducing training time, and improving results in a wide range of tasks, like the introduction of semantic information as organized color data [52,53], sampling and registration [54], compression [55], image enhancement and dehazing [56,57], spectral unmixing [58], and soft-labeling [59].

In addition to ABANICCO’s proven success in automatic color classification, naming, and segmentation, its efficiency is highlighted by its computational simplicity, allowing it to run quickly even on low-end computers. Our experiments show that ABANICCO was able to analyze the 380 × 570 pixels image in Figure 4 in less than a second on a mid-end computer (an HP laptop with i5, 2.4 Hz, and 16Gb RAM).

While the proposed method has many strengths, it does have a drawback that we aim to address in future work: the pipeline for image segmentation currently requires manual intervention by the user, specifically in the class remodeling step. Our future work will explore integrating various automatic clustering methods, to automate the process entirely. With this improvement, ABANICCO will become a fully automatic, non-supervised, unbiased option for image segmentation, including video analysis, without relying on human intervention.

## 7. Conclusions

In this paper, we have proposed a novel method for color classification, naming, and segmentation, called ABANICCO. Conventional color analysis techniques can be biased, due to human perception and context. By contrast, our method is solely based on color theory, space geometry, fuzzy logic concepts, and multi-label systems.

ABANICCO automatically detects the colors present in an image, classifies them according to color theory, and shows how they are organized, using an innovative polar AB color space. Each detected color is then assigned a multi-label name according to irregular membership functions, considering the differences in color perception, successfully bridging the semantic gap. Finally, the classification output of ABANICCO can be used directly for either pure color segmentation or, with the simple step of boundary remodeling, for controlled, informed image segmentation based on color theory.

We evaluated ABANICCO in two phases. Firstly, we assessed its accuracy in color classification and naming, using Level 2 of the ISSC–NBS color classification system, and achieving perfect accuracy. Next, we extended ABANICCO’s capabilities to the problem of image segmentation through a simple pipeline, comparing its performance to the state-of-the-art fuzzy segmentation and color-based object segmentation methods, proving that it outperforms other methods, while being much more straightforward and efficient.

ABANICCO represents a substantial advancement in the field of color analysis, and has the potential to be applied in a broad range of real-world applications.

## Figures and Tables

**Figure 1 sensors-23-03338-f001:**
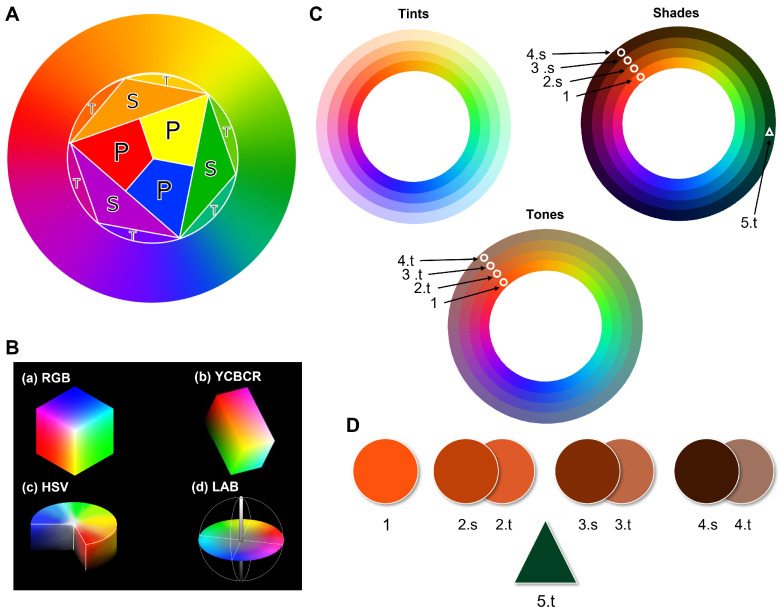
Color theory bases: (**A**) representation of color wheel; the color wheel is obtained from the primary colors (P) Red, Yellow, and Blue; by blending the primary colors, we get the secondary (S) colors, Green, Purple, and Orange; finally, by blending the secondary colors, we obtain the tertiary (T) colors, Yellow-Green, Blue-Green, Blue-Violet, and Red-Violet; (**B**) depiction of the concept of tints, shades, and tones with the color wheel in A; in every wheel, each ring going outward shows a 25% increase in added white for tints, black for shades, and gray for tones; these show the near-achromatic colors. (**C**) the four main color spaces used in digital imaging—RGB, YCbCr, HSV, and CIELAB; (**D**) color picker results displaying the colors within the shapes marked in the shades (s)- and tones (t)-modified color wheels; the circles show the effect of adding black (shades) or gray (tones) to orange; the triangle shows a dark shade of green called forest green; the obtained results show that what we understand as brown is not a pure color but rather shades of pure colors ranging from red to warm yellow.

**Figure 2 sensors-23-03338-f002:**
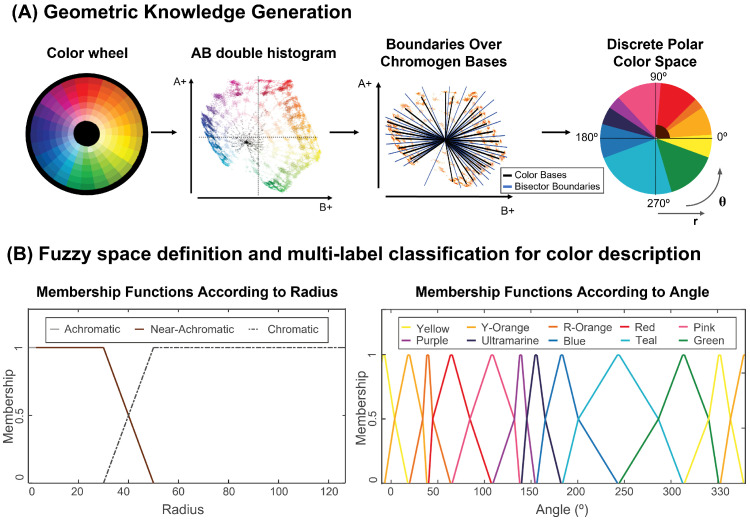
The two main steps of ABANICCO: (**A**) geometric knowledge generation; we used color theory to identify in the color wheel the localization of the different pure hues, shades, and tints within the reduced AB space of CIELAB; applying geometry, we used bisectors as the best boundaries between pure hues; by these means, we obtained a discrete polar color space divided into 12 different color categories—Pink, Red, Red-Orange, Yellow-Orange, Yellow, Green, Teal, Blue, Ultramarine, Purple, Brown, and Achromatic; the first 10 categories described hue, depending on the angle; the last two described chromaticity, and depended on the radius; (**B**) fuzzy space definition and multi-label classification for color description; we formalized fuzzy rules to translate the obtained discrete color polar space back into a continuous space that better mirrored human linguistics; based on the bisectors employed for the discretization of the space, we were able to define areas of absolute chromogen certainty and gradients of membership to the adjacent colors on both sides, which was done for both the radius (accounting for chromaticity) and the angle (accounting for hue); finally, with color theory, fuzzy membership functions, and multi-label classification, we assigned multiple, non-exclusive labels to each detected color, for accurate naming and description.

**Figure 3 sensors-23-03338-f003:**
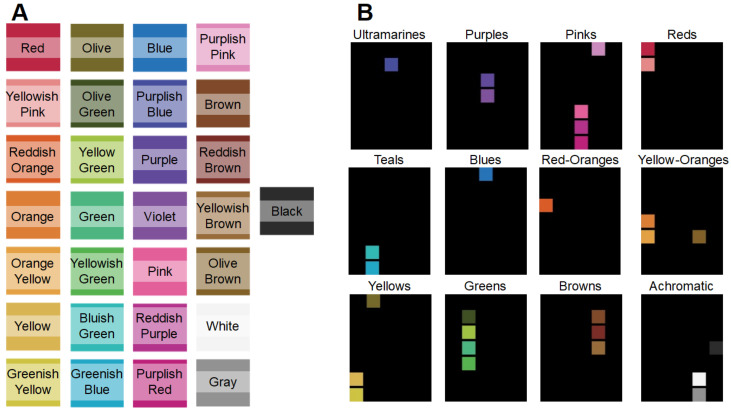
Evaluation of our method with ISCC–NBS system of color categories: (**A**) Level 2 of the ISCC–NBS system, with each color’s name superimposed over its corresponding square; (**B**) ABANICCO classification of image (**A**) into the resulting 12 categories.

**Figure 4 sensors-23-03338-f004:**
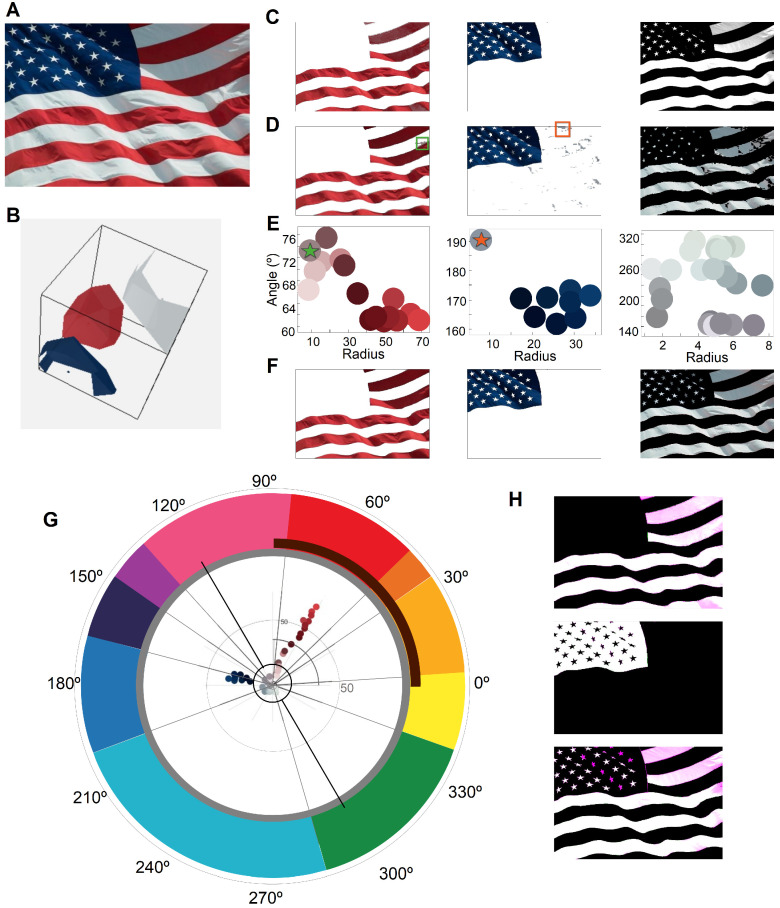
Comparison of segmentation by our method (ABANICCO) versus fuzzy method in [41]: (**A**) original natural image with a flag waving; (**B**) fuzzy color space built from extracted prototypes, using the fuzzy method; (**C**) row of results (red, blue, and white mapping) of the segmentation, using supervised prototypes with the proposal in [41]; this technique used membership degrees to represent the segmentation; thus, lower saturation corresponded to lower membership certainty; (**D**) row of results (red, blue, and white category) of the segmentation using ABANICCO; the green and orange squares show areas where our method failed to segment the stripes accurately; (**E**) colors within the red, blue, and light categories found by ABANICCO; the stars mark the color of the areas of failed segmentation in (**D**); the green (orange) star marks the color of the area within the green (orange) rectangle; (**F**) row of segmentation results using ABANICCO after adjusting the boundaries; (**G**) polar description of the image with the original boundaries in gray and the revised boundaries shown with dotted black lines; (**H**) overlap of the masks obtained in row F and those obtained by [42]; the different shades of pink indicate where the method in [42] output an uncertain membership to that class—the stronger the pink, the lower the membership. In these images, we can see how their method fails in areas where the differences in illumination changed the expected colors: dark red (brown) and dark white (gray).

**Figure 5 sensors-23-03338-f005:**
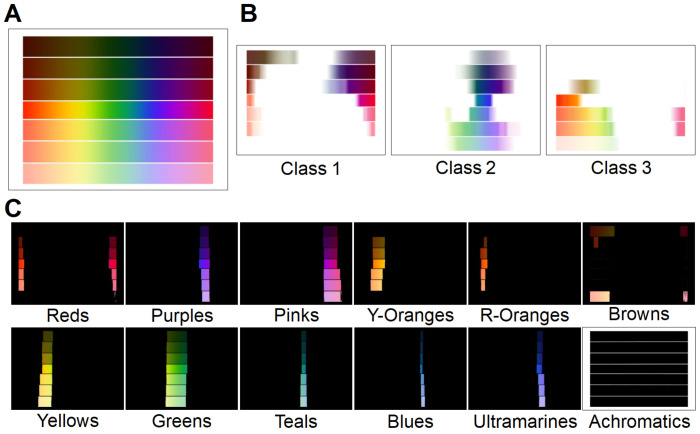
Comparison of segmentation with our method to fuzzy method in [41], using a simplified version of the image shown in Figure 1: (**A**) the middle rectangle shows the unraveled color wheel; the rectangles above (below) add an increasing quantity of black (gray) to represent shades (tones); (**B**) results of classification using fuzzy color spaces with three arbitrary classes; the lack of labels, and the gradual transitions between colors, resulted in membership maps with small areas of absolute membership, and unclear class separations; (**C**) results of color classification using ABANICCO; our method automatically divided the image into 12 classes with clear boundaries, corresponding to the main hues present plus the near-achromatic brown shades and tones and the achromatic white and black (empty for this particular image).

**Figure 6 sensors-23-03338-f006:**
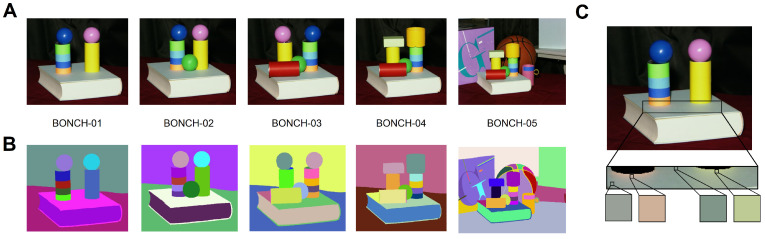
The ITTP-Close experimental dataset: (**A**) RGB images present in the dataset; (**B**) the corresponding groundtruths; (**C**) a closer look at the different colors in one area, as defined by the ground truth.

**Figure 7 sensors-23-03338-f007:**
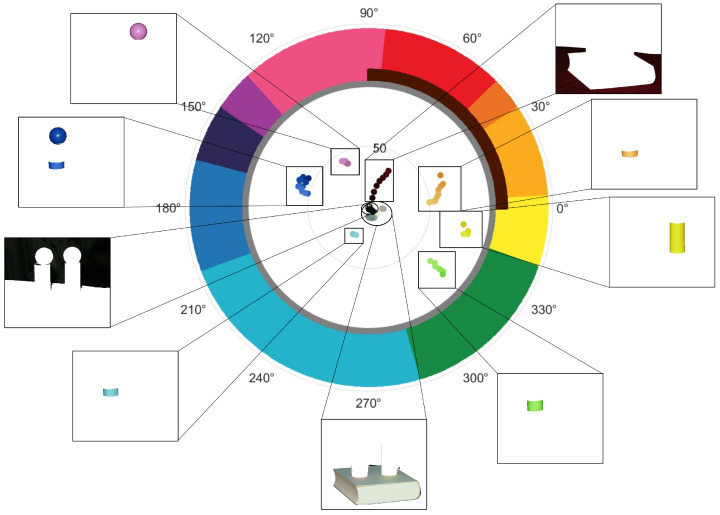
Segmentation of sample image BONCH-01 of the ITTP-Close experimental dataset, using class remodeling of ABANICCO’s detected colors as clustering-based image segmentation.

**Table 1 sensors-23-03338-t001:** Full analysis of each shade of the official ISCC–NBS color classification Level 2.

ISCC–NBS System Name	Algorithm’s Classification	Algorithm’s Full Description
Red (R)	Red	86.09% Red and 13.91% Pink
Yellowish Pink (Ypk)	**Red**	58.77% Brown, 40.04% Red, and 1.19% Pink
Reddish Orange (Ro)	Deep Orange	78.67% Deep Orange and 21.33% Red
Orange (O)	Light Orange	62.69% Light Orange and 37.31% Deep Orange
Orange Yellow (OY)	Light Orange	93.41% Light Orange and 6.59% Yellow
Yellow (Y)	Yellow	55.69% Yellow and 44.31% Light Orange
Greenish Yellow (Gy)	Yellow	97.24% Yellow and 2.76% Green
Olive (Ol)	Yellow	97.61% Yellow and 2.39% Light Orange
Olive Green (Olgr)	Green	74.38% Green and 25.62% Yellow
Yellow-green (YG)	Green	64.75% Green and 35.245% Yellow
Green (G)	Green	62.73% Green and 37.27% Teal
Yellowish Green (Yg)	Green	98.27% Green and 1.73% Teal
Bluish Green (Bg)	Teal	86.26% Teal and 13.71% Green
Greenish Blue (Gb)	Teal	75.88% Teal and 24.12% Blue
Blue (B)	Blue	88.12% Blue and 11.78% Ultramarine
Purplish Blue (Pb)	Ultramarine	100% Ultramarine
Violet (V)	Purple	57.27% Purple and 42.73% Ultramarine
Purple (P)	Purple	73.47% Purple and 26.53% Pink
Pink (P)	Pink	74.31% Pink and 25.69% Red
Reddish Purple (Rp)	**Pink**	100% Pink
Purplish Red (Pr)	Pink	82.42% Pink and 17.58% Red
Purplish Pink (Ppk)	Pink	85.01% Pink and 14.99% Purple
Brown (Br)	Brown	77.72% Brown, 7.16% Light Orange, and 15.12% Deep Orange
Reddish Brown (Rbr)	Brown	50.10% Brown, 8.28% Deep Orange, and 41.62% Red
Yellowish Brown (Ybr)	Brown	69% Brown and 31% Light Orange
Olive Brown (Olbr)	Brown	64.40% Brown, 26% Light Orange, and 9.60% Yellow
White (Wh)	Neutral	67.50% Yellow and 32.50% Light Orange
Gray (Gy)	Neutral	67.50% Yellow and 32.50% Light Orange
Black (Bk)	Neutral	67.50% Yellow and 32.50% Light Orange

## Data Availability

All the data, code, resources, and additional materials used in our research are available in https://github.com/lauranicolass/ABANICCO (accessed on 19 March 2023).

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
