# Peer review of "ABANICCO: A New Color Space for Multi-Label Pixel Classification and Color Analysis"

_sensors, 2023, doi:10.3390/s23063338_

Round 1

Reviewer 1 Report

This paper presents a New Colour Space for Multi-Label Pixel Classification and Colour Analysis. It is an interesting research. 

1. What is the relationship between geometric analysis, colour theory, fuzzy colour spaces, and multi-label systems?  If there is a general diagram,the novel method can be more understandable.

2. Can more quantitative precision analysis be added in experiment part? 

3. Some detials about format and punctuation, such as Figure 1 on page 3.

Author Response

Dear Reviewer 1,

We would like to express our sincere gratitude for your valuable comments and feedback that have been incredibly helpful in improving the quality of our paper. We have carefully considered all your suggestions and have implemented them accordingly, improving the explanations of the research design, methods, presentation of results, and conclusions. These changes to the original manuscript are written in red for easier locating.

COMMENT 1: What is the relationship between geometric analysis, colour theory, fuzzy colour spaces, and multi-label systems? If there is a general diagram, the novel method can be more understandable.

Answer 1: Each of these disciplines contributes in different ways to the development of ABANICCO. 

Colour theory is the basis of ABANICCO. It allows us to find the colour bases and name the detected colours accordingly without committing misnomers like “bluish-brown”. It also eliminates human bias with colour perception and detection and reduces the effect of noise and changes in illumination.

Geometric analysis is used to find the appropriate boundaries between the colour bases, as presented in Section 5.1. In the Supplementary material, we show that bisectors far exceed the performance of machine learning methods.

Fuzzy colour space is a term used to describe a collection of fuzzy colours corresponding to the colour categories employed in a certain context/application and/or for a specific user. This term was popularised by J.Chamorro et al. in the papers we reference as [39-42,  48-49]. Inspired by their work, we use the principles of fuzzy theory to translate the created discrete AB colourspace back into a continuous colour space for naming and description. Fuzzy theory is the best approach for modelling colours because it mirrors the lack of clear boundaries between them. We take the located colour bases and the bisector boundaries and create membership functions under fuzzy theory. 

Finally, we use multilabel classification to, from those membership functions, create accurate descriptions of the detected colours in a way that is the most similar to human linguistic while still serving as a digital mapping.

To better reflect this, we have improved the graphical abstract in the supplementary material and included further explanations at the beginning and end of the Methods section and in Figure 2. We include here the new paragraph added to the end of the Methods section:

To summarize, we have made use of concepts from different disciplines in the development of ABANICCO. First, we have centred our model on colour theory, which we have used to find the location of colours in space and represent and name them accordingly. Additionally, we have employed geometric analysis to find the best boundaries between colour bases. Finally, we have combined fuzzy theory and multilabel classification to create a continuous space. This resulting space is closer to human understanding of colour from which non-exclusive labels can be assigned to each detected colour to name them accurately, thus bridging the semantic gap.

COMMENT 2: Can more quantitative precision analysis be added to the experiment part?

Answer 2: This is the part of our work where we struggled the most. We found it extremely challenging to perform a proper quantitative analysis of ABANICCO. There are several reasons why it can be challenging to have a more precise quantitative assessment of colour analysis models:

  • Subjectivity: colour perception is subjective and can vary from person to person. Therefore, it can be challenging to create a standardised method of colour assessment that is applicable to everyone.
  • Complexity: colour analysis models are often complex and can involve a range of factors, such as colour spaces and colour gamuts, and colour profiles. It is difficult to compare the performance of models that treat colours so differently.  
  • Lack of data: There is a lack of comprehensive and standardised labelled datasets that accurately represent the full range of colours that colour analysis models need to account for. Human-made annotations are inevitably subjective and often consider colour as an attribute of objects rather than as its own entity like ABANICCO does.

Overall, quantifying the precision and efficiency of colour analysis models requires a multi-disciplinary approach, which is what we have done with our different sets of experiments. We have shown ABANICCO classification and naming performance by comparing it to a standardised dataset,  while segmentation performance is compared to two different state-of-the-art colour segmentation methods.

The experiments in sections 6.1 and 6.2.3. are quantitative experiments, with numerical results giving a clear description of the accuracy of ABANICCO for colour classification and naming in the former and for object segmentation in the latter. 

However, we decided not to pursue quantitative comparison in the experiments of sections 6.2.1. and 6.2.2 because the output of the state of the art method is entirely different, making it difficult to achieve a relevant metric that compares it to ABANICCO. With their approach, Mengíbar-Rodríguez et al. [41] obtain a membership map for each category, with 1 (white)  being absolute certainty of belongingness and 0 (black) being absolute certainty of not belongingness. While we could have obtained numerical metrics like the Dice coefficient by comparing our segmenting output with theirs, it would have been a poorly chosen metric that would not consider the differences certainty present over their maps. Therefore, for lack of adequate, problem-aware, quantitative metrics [50], we preferred to assess the comparison qualitatively in Figures 4 and 5.

To better reflect these ideas, we have rewritten parts of section 6, and we have included a new paragraph at the start of the section to reflect these difficulties better:

While colour analysis is a crucial component in various computer vision tasks, the assessment of analytic models is decidedly troublesome. As explained in Section 2.1, the perception of colour is influenced by many subjective factors, such as the individual's age, gender, culture, and personal experiences. While most computer vision algorithms are assessed using human-made labels, in the case of colour analysis, these human-made annotations can never be objective. They would differ depending on the group creating them. In addition, colour measurement devices can be influenced by factors such as lighting conditions, substrate colour, and sample size, making it difficult to achieve consistent and reliable results. Moreover, in most applications, colour is often considered an attribute of an object rather than a standalone concept. This distinction, along with the challenges posed by human perception, context, and interpretation, results in a scarcity of ground truth data and adequate analysis metrics for colour analysis as an individual concept. 

To address these difficulties, we have designed two types of experiments. The first type of experiment quantitatively evaluates ABANICCO’s accuracy in colour classification and colour naming using as benchmark the ISCC NBS colour classification Level 2 (Section 6.1). The second class of experiments explores the use of ABANICCO in colour-based image segmentation ( Section 6.2). We first qualitatively evaluate  the performance of ABANICCO’s clustering-based colour segmentation pipeline against the state-of-the-art fuzzy-based colour segmentation algorithm ( Sections 6.2.1 and 6.2.2) and finally, we quantitatively assess the validity of this pipeline for object segmentation using the ITTP-close dataset (Section 6.2.3). Thanks to this empirical evaluation, we provide a numerical comparison of the proposed model against state-of-the-art benchmarks.

COMMENT 3: Some details about format and punctuation, such as Figure 1 on page 3.

Answer 3: We have revised the entirety of the document and improved its format and punctuation, with special attention to all the figures.

Once again, we thank you for your time and effort in reviewing our paper. Your feedback has been invaluable in improving the quality of our work.

Reviewer 2 Report

abanicco: Use labeled pixel color space"... This study proposes a clear and practical method of color classification to improve the quality of this manuscript. Notes and issues that need to be considered/more correct.

1. Highlight the highlights and characteristics of this study.

2. Please compare this method with the shortcomings of existing color classification methods.

Author Response

Dear Reviewer 2,

We would like to express our sincere gratitude for your valuable comments and feedback on our paper, which have been helpful in improving the overall quality. We have carefully considered all your suggestions and have implemented them accordingly. These changes to the original manuscript are written in blue for easier locating.

COMMENT 2: Highlight the highlights and characteristics of this study.

Answer 1: We thank the reviewer for this suggestion. We have included the highlights of our work at the end of Section 1, which certainly improves the clarity of our paper. We copy these highlights here:

The main highlights of our method are:

  • ABANICCO is an easy-to-use, automatic, non-supervised method for colour analysis segmentation, classification, and naming.
  • It is based on objective colour theory, thus eliminating human colour bias and making the analysis trustworthy and reproducible.
  • It incorporates fuzzy logic concepts to approximate colour naming to human linguistics and understanding, thus bridging the semantic gap.
  • Finally, ABANICCO allows the user to make informed decisions and modify the obtained results to fit specific applications. This is possible thanks to ABANICCO's ease of use and excellent graphical representation of the detected colours.

COMMENT 2: Please compare this method with the shortcomings of existing colour classification methods.

Answer 2: Each method has its own shortcomings. The main issues common to most colour analysis models are:

  • High complexity: Most models nowadays rely on complicated architectures and heavy training with a high amount of data. This makes these models hard to use and inaccessible for anyone without the same knowledge as the original researchers. ABANICCO is the complete opposite. We have created a model based on the simplest foundations of other disciplines, making its process easy to understand. Moreover, we provide an accessible graphical representation of the colours that allows the user to make informed decisions and generate their own analysis. 
  • Lack of standardisation and generalizability: Because most models rely on heavy training, it is difficult to use them for new applications. Moreover, it is difficult to compare their performance among different users. Since ABANICCO is automatic and does not need annotations, it can be applied to any task. Moreover, its ease of use and understanding makes it easy to change any necessary aspects while describing them for easy standardisation and comparison between users.  
  • Limited training data: Many colour classification methods require large amounts of training data to classify objects accurately. However, obtaining and labelling this data can be time-consuming and expensive. Moreover, because colour perception is subjective, these annotations can never be fully accurate and are heavily affected by lighting, optical phenomena and personal bias. Because ABANICCO does not need annotations, it is much more objective in its analysis. 

We have included a new paragraph in the Discussion section to illustrate these differences between ABANICCO and other state-of-the-art methods:

Thus, ABANICCO successfully deals with the common pitfalls of other colour analysis approaches. By basing our model on objective colour theory, we avoid the danger of subjectivity, inability to handle variations in lighting and noise, and lack of standardisation of other models. Moreover, we ensure that the range of detectable colours is maximum by building ABANICCO from CIELAB colour space. Finally, ABANICCO is the least complex model compared to the state-of-the-art. This, together with not needing any annotations, makes ABANICCO practical for a wide range of applications and accessible to researchers regardless of their field.

Round 2

Reviewer 1 Report

The revised paper is more understandable for readers. We think the article can be accepted.